# Combinatorial Effect of Cold Atmosphere Plasma (CAP) and the Anticancer Drug Cisplatin on Oral Squamous Cell Cancer Therapy

**DOI:** 10.3390/ijms21207646

**Published:** 2020-10-15

**Authors:** Chang-Min Lee, Young-IL Jeong, Min-Suk Kook, Byung-Hoon Kim

**Affiliations:** 1Department of Dental Materials, School of Dentistry, Chosun University, Gwangju 61452, Korea; ckdals1924@daum.net (C.-M.L.); jeongyi@chosun.ac.kr (Y.-I.J.); 2Department of Maxillofacial Surgery, School of Dentistry, Chonnam National University, Gwangju 61186, Korea

**Keywords:** cold atmospheric plasma, reactive oxygen species, squamous cell carcinoma, cisplatin, combination therapy

## Abstract

Cold atmospheric plasma (CAP) has been extensively investigated in the local treatment of cancer due to its potential of reactive oxygen species (ROS) generation in biological systems. In this study, we examined the synergistic effect of combination of CAP and cisplatin-mediated chemotherapy of oral squamous cell carcinoma (OSCC) in vitro. SCC-15 OSCC cells and human gingival fibroblasts (HGF-1) cells were treated with cisplatin, and then, the cells were irradiated with CAP. Following this, viability and apoptosis behavior of the cells were investigated. The viability of SCC-15 cells was inhibited by cisplatin with a dose-dependent manner and CAP treatment time. HGF-1 cells also showed decreased viability by treatment with cisplatin and CAP. Combination of 1 μM cisplatin plus 3 min of CAP treatment or 3 μM cisplatin plus 1 min of CAP treatment showed a synergistic anticancer effect with appropriate cytotoxicity against normal cells. ROS generation and dead cell staining were also increased by the increase in CAP treatment time. Furthermore, tumor-suppressor proteins and apoptosis-related enzymes also increased according to the treatment time of CAP. We showed the synergistic effect of cisplatin and CAP treatment against SCC-15 cells with low cytotoxicity against normal cells.

## 1. Introduction

Cold atmospheric plasma (CAP) has been extensively studied in the biomedical field in recent decades due to its biological and physicochemical properties [1,2,3,4]. Notably, CAP has recently been spotlighted in the local treatment of cancer because CAP in the biological system generates reactive oxygen species (ROS) and/or reactive nitrogen species (RNS), i.e., excessive ROS or RNS generation by CAP treatment induces oxidative stress against cancer cells and subsequently results in apoptotic death [5,6,7,8]. These unique properties of CAP are regarded as a safe treatment option for cancer patients because it can be applied to the specific body site of action without systemic side-effects against the human body [9,10,11,12]. For example, Kurake et al. reported that hydrogen peroxide and nitrous ion in the plasma-irradiated culture medium selectively induced the apoptotic death of glioblastoma tumor cells, but they did not significantly affect to the viability of normal mammary epithelial cells [7]. ROS/RNS generated by CAP treatment triggers molecular signaling pathways and promotes mitochondrial perturbation, leading to apoptosis [8]. Otherwise, plasma-activated medium (PAM) is known to induce immunogenic cell death by activation of the innate immune system [13]. Furthermore, Shi et al. reported that CAP treatment against cancer cells induces enriched p53 pathway in SCC-15 cells [14]. Despite these advantages of CAP treatment, some drawbacks have been reported [15,16,17]. For example, CAP treatment aroused focal mucosal erosion with superficial ulceration and necrosis with inflammatory reaction, even though these mucosal damages healed within a few days [15]. Since the free radicals induced by non-thermal plasma (NTP) have the danger of the hemolytic activity against red blood cells, the mild condition of ROS/RNS induction by CAP is required for safe treatment on human diseases [16]. Furthermore, cellular mutagenic changes induced by CAP treatment should be also considered to find precise conditions for safe treatment modality [17].

Otherwise, the efficacy of cancer cell treatment with CAP can be improved by co-treatment with gold nanoparticles, since gold nanoparticles have a synergistic effect on the production of ROS/RNS in the treatment of the cancer cells by the NTP system [18,19]. Moniruzzaman et al. reported that co-treatment of CAP and X-irradiation using sulfasalazine generates hydroxyl radicals 24 times higher than X-irradiation only, and then accelerates apoptotic death of human leukemia Molt-4 cells [20]. Furthermore, Lee et al. reported that CAP treatment sensitizes tamoxifen-resistant MCF-7 cells against tamoxifen treatment and overcomes drug resistance [21]. The efficacy of CAP should be improved to maximize the curative value of human diseases with minimized side-effects against normal tissues.

Cisplatin, which is a metal-based anticancer drug, has been extensively used as a chemotherapy medication for various kinds of cancers such as ovarian cancer, head and neck cancer, cervical cancer, breast cancer, bladder cancer, esophageal cancer, lung cancer, brain tumors and neuroblastoma [22,23,24,25,26,27,28,29]. Cisplatin works by interfering with DNA replication and inhibits the growth of cancer cells [22]. Although cisplatin has been used for cancer patients for several decades, severe side-effects such as bone marrow suppression, hearing problem, electrolyte disorders, nephrotoxicity and cardiotoxicity still inhibit its clinical application and degrade the quality of patients’ lives [30,31,32]. From these points of view, various regimens have been investigated to overcome these drawbacks [33,34,35,36]. For example, the electroporation significantly improves anticancer activity of cisplatin against metastatic pancreatic cancer [33,34]. Short electric pulses against cancer cells increase transitional permeabilization of the drug molecule through the cellular membrane and improve anticancer activity. Furthermore, polymeric micelles or nanoparticles incorporating cisplatin significantly improve the anticancer activity of cisplatin against cancer cells [35,36]. Nishiyama et al. reported that polymeric micelles remarkably improve blood circulation of cisplatin and effectively accumulated in a solid tumor following complete regression of the solid tumor [35].

In this study, we investigated the combinatorial effect of CAP and cisplatin against squamous cell carcinoma cells in vitro. Since cisplatin also induces oxidative damage both in normal cells and cancer cells, oxidative stress can be maximized by the combination of CAP and cisplatin against cancer cells. Furthermore, this treatment regimen may reduce the adverse effects against normal cells through local treatment of cancer cells. We studied the synergistic effect of the CAP/cisplatin combination on ROS generation, apoptosis and anticancer activity using squamous cell carcinoma cells and normal cells.

## 2. Results and Discussion

The ROS producing capacity of CAP is known to induce various biological activities [1,2,3,4,5]. The ROS/RNS generated in the biological system by CAP can be applied for specific killing of cancer cells. Notably, CAP is regarded as a promising candidate for local irradiation of superficial carcinoma such as squamous cell carcinoma because CAP application in cancer is limited to top cell layers, i.e., the effective depth of tissue penetration of CAP treatment is less than 100 μm [37]. Otherwise, the delivery of anticancer drugs via systemic administration against superficial carcinoma such as skin cancer and oral squamous cell carcinoma has limitations in clinical applications, and very low fractions of administered drug can be reached in the tumor region [38,39]. These delivery problems of anticancer drugs such as cisplatin amplify various adverse effects on the human body [28,29,30,31,32,33,34]. From these points of view, we designed a combination of CAP treatment and cisplatin to amplify the anticancer activity of cisplatin and to minimize the adverse effects against normal cells.

### 2.1. Optical Emission Spectroscopy (OES) Analysis

For the treatment of cancer cells using CAP, the distance between the nozzle and media in a 12-well plate was adjusted to 3 cm, and the flow rate of Ar gas was adjusted to 5 L/min (Figure 1A,B).

For analysis of the plasma jet plume from CAP devices, OES was employed, and the measurement result is shown in Figure 2. This result indicated that the specific peaks of Ar^+^, OH and O^-^ ions were observed in the UV range (200 nm–400 nm) and visible range (690 nm–950 nm). Furthermore, this result indicates that CAP equipment in this study successively produces a plasma jet from Ar gas. Many scientists reported the curative efficacy of Ar plasma jets against various disease models, including wound healing, bacterial infection and cancer [40,41,42]. Cheng et al. reported that the Ar plasma jet efficiently decreases new blood vessels in streptozotocin-induced diabetic rats [40]. Furthermore, Lou et al. also reported that the Ar plasma jet efficiently destroys the bacteria cell wall and then induces successful killing of bacteria [41].

### 2.2. Single Treatment of Cisplatin or CAP against HGF-1 and SCC-15 Cells

The effect of sole treatment of cisplatin or CAP against SCC-15 cells and HGF-1 cells was examined in vitro, as shown in Figure 3. The viability of SCC-15 cells and HGF-1 cells was gradually decreased according to the increase in cisplatin concentration and CAP treatment time. Interestingly, SCC-15 cells showed lower viability in the sole treatment of cisplatin compared to HGF-1 cells, as shown in Figure 3A. That is, the viability of SCC-15 cells was lower than 50% at 3 μM cisplatin, while HGF-1 cells showed higher than 90% in cell viability at the same concentration. Moreover, SCC-15 cells resulted in lower viability by CAP treatment compared to HGF-1 cells (Figure 3B). As shown in Figure 3B, the viability of SCC-15 cells was less than 50% at a treatment time of 3 min (180 s), while more than 70% of HGF-1 cells were viable at the same concentration. These results indicated that HGF-1 cells properly tolerated the treatment of cisplatin or CAP when compared with SCC-15 cells. As shown in Table 1, the IC_50_ values of HGF-1 and SCC-15 cells by treatment of cisplatin were 16.0 μM and 2.8 μM, respectively. Furthermore, the viability of SCC-15 cells was also gradually decreased according to the CAP treatment time. As shown in Table 1, the IC_50_ value of HGF-1 and SCC-15 cells by treatment of CAP was approximately higher than 300 s and 173 s, respectively. Interestingly, the IC_50_ value of HGF-1 cells was higher than that of SCC-15 cells.

### 2.3. Combinatorial Treatment of Cisplatin and CAP against HGF-1 and SCC-15 Cells

From the results of the viability of cells, the optimal combination of cisplatin concentration and CAP treatment time was studied. As shown in Figure 4, 1 μM or 3 μM of cisplatin was combined with various treatment times of CAP to find the maximal anticancer activity against cancer cells and the minimal cytotoxicity against normal cells. Practically, higher than 80% of HGF-1 cells was viable at 5.0 μM (Figure 3A). As shown in Figure 4, the cells responded to CAP treatment time, and the viability of cells gradually decreased according to the increase in CAP treatment time when the various treatment times of CAP were combined with 1 μM or 3 μM cisplatin. Even though the viability of HGF-1 cells also decreased according to the treatment time of CAP, their viability was higher than those of SCC-15 cells at all treatment regimens. When 1 min treatment of CAP was combined with 1 μM cisplatin, less than 60% of SCC15 cells were viable, while more than 80% of HGF-1 cells were viable at same treatment regimen. One-minute treatment of CAP combined with 3 μM cisplatin induced a decrease in viability of SCC-15 cells of less than 40%, while more than 70% of HGF-1 cells were viable at same treatment modality. Furthermore, the viability of HGF-1 cells at 3 μM cisplatin was not significantly changed compared to the results of 1 μM cisplatin, while the viability of SCC-15 cells was significantly decreased at 3 μM cisplatin compared to 1 μM cisplatin. These results indicated that SCC-15 cells are more sensitive to cisplatin treatment compared to HGF-1 cells. Even though the higher dose of cisplatin in combination with CAP resulted in stronger antitumor activity, the lower dose of cisplatin combined with the longer treatment time of CAP is preferred for the treatment of epithelial cancer, because CAP brings only about local irritations in the normal tissues, while cisplatin induces systemic side-effects against the human body.

Figure 5 also supports the results of Figure 3 and Figure 4. Even though the morphology and the density of HGF-1 cells were changed at 3 μM cisplatin plus 3 min of CAP treatment, the cell density and the morphology of HGF-1 cells were relatively maintained at 1 μM cisplatin, 1 μM cisplatin plus 3 min of CAP treatment and 3 μM cisplatin, as shown in Figure 5A. Otherwise, SCC-15 cells showed greater changes in the morphology and the cell density by treatment of cisplatin and/or CAP. At 3 μM cisplatin plus 3 min of CAP treatment, the cell density and the morphology of SCC-15 cells were significantly decreased and changed, as shown in Figure 5B. These results supported the results of Figure 4. Pasqual-Melo et al. also reported the anticancer efficacy of plasma jets using a kINPen argon plasma jet [42]. In their report, the plasma jet revealed higher cytotoxicity in malignant cells such as SCC-13 and A431 cells but not in non-malignant HaCaT keratinocytes. We also obtained similar results from Ar plasma treatment against non-malignant HGF-1 cells and malignant SCC-15 cells, as shown in Figure 4 and Figure 5.

The live/dead cell staining of SCC-15 cells supported the results of Figure 4 and Figure 5. As shown in Figure 6, dead cells with a red color was gradually increased with the combination of cisplatin and CAP treatment compared to control treatment. The higher treatment time of CAP significantly decreased the live cells, while dead cells simultaneously increased compared to control or cisplatin only. Otherwise, the live cell staining in HGF-1 cells with a green color was relatively higher than dead cell staining (Appendix A), indicating that CAP treatment efficiently influences the viability of SCC-15 cells rather than HGF-1 cells. Figure 4, Figure 5 and Figure 6 show that the combination of cisplatin and CAP treatment induces the synergistic death of cancer cells with alleviated cytotoxicity against normal cells.

### 2.4. ROS Generation and Apoptosis/Necrosis Signals

Figure 7 shows the ROS generation from HGF-1 cells and SCC-15 cells by treatment with cisplatin and CAP. As expected, the higher CAP treatment time induced increased ROS production in both HGF-1 cells (Figure 7A,B) and SCC-15 cells (Figure 7C,D). Interestingly, ROS production of HGF-1 cells was not significantly changed between 1 μM and 3 μM cisplatin, while the ROS level was significantly changed in SCC-15 cells, as shown in Figure 7C,D. Notably, 1 μM cisplatin plus 1 min CAP or 3 μM cisplatin plus 1 min CAP resulted in higher than 300% or 500% increases in ROS levels in SCC-15 cells, respectively, while ROS generation in HGF-1 cells in same treatment option was controlled under 150% and 170%, respectively. These results indicate that CAP treatment against SCC-15 cancer cells is an efficient candidate for the generation of ROS and specific suppression of cancer cells.

Figure 8 shows the expression of apoptosis-related signals such as PTEN, cleaved caspase-9 (Cas-9) and p53 proteins of SCC-15 cells and HGF-1 cells. As shown in Figure 8, tumor-suppressor protein PTEN was gradually increased both in SCC-15 cells and HGF-1 cells, indicating that CAP treatment increases anti-tumor activity with expression of tumor-suppressor genes. Furthermore, apoptosis-related enzyme Cas-9 in SCC-15 cells was remarkably changed according to the CAP treatment of both 1 μM and 3 μM cisplatin, i.e., the higher treatment time of CAP gradually increases the expression intensity of Cas-9. Otherwise, the changes in Cas-9 expression intensity in HGF-1 cells were relatively not significant compared to SCC-15 cells. The expression intensity of p53 protein was also gradually increased by treatment with CAP in both SCC-15 cells and HGF-1 cells. These results indicated that CAP treatment in combination with cisplatin induces apoptosis of tumor cells and increases the tumor-suppressor protein, resulting in an increase in antitumor activity. Figure 9 shows the flow cytometric analysis of apoptosis/necrosis of SCC-15 cells. Apoptosis/necrosis of SCC-15 cells was gradually increased according to the increase in treatment time of CAP. Furthermore, the early/late apoptosis of cancer cells at 3 μM of cisplatin combined with CAP treatment was higher than 1 μM of cisplatin. Notably, late apoptosis was relatively higher at 3 min CAP treatment, while early apoptosis was evidently higher than late apoptosis at less than 2 min CAP treatment. With sole treatment of cisplatin, early apoptosis was dominant at 3 μM of cisplatin rather than late apoptosis, while its value at 1 μM of cisplatin was lower than late apoptosis. These results might be due to the fact that SCC-15 cells respond to cisplatin treatment during the initial 6 h, and then apoptosis/necrosis is accelerated following cell death. Xu et al. also reported that CAP increased caspase-9 activity of multiple myeloma and induced apoptosis/necrosis of cells [43]. Saadati et al. also reported that indirect/direct treatment of B16F10 melanoma cells using CAP induces apoptosis-related protein expression, such as BAX/BCL-2 [44]. They reported that CAP treatment greatly inhibited the viability of malignant cells in vitro and in vivo through apoptosis.

The results of apoptosis analysis of cells indicated that CAP treatment efficiently induces apoptotic death of cancer cells. Kurake et al. also reported that plasma-activated medium (PAM) selectively killed glioblastoma cells but not in normal mammary epithelial cells [7]. They reported that ROS such as hydrogen peroxide (H_2_O_2_) and nitrous ion (NO_2_^−^) were detected in the plasma-irradiated culture medium, and these components contributed to inducing apoptotic death of glioblastoma tumor cells, while the viability of normal mammary epithelial cells was properly maintained. Furthermore, Ikeda et al. also reported that PAM shows anticancer activity through inhibition of cancer-initiating cells [45]. They observed that PAM selectively induces apoptotic death of cancer cells but not in normal cells. Furthermore, they also obtained positive results from PAM/cisplatin combination therapy, i.e., the combination of PAM and cisplatin has synergistic cancer cell-killing activity when compared to PAM or cisplatin only. Our results also indicated that anticancer activity against SCC-15 cells can be emphasized by the combination of CAP and cisplatin. Notably, this combinatorial treatment dominantly inhibited viability of malignant SCC-15 cells with reduced cytotoxicity against non-malignant HGF-1 cells. Even though many scientists investigated the anticancer efficacy of CAP, we discovered the optimal treatment modality of the cisplatin/CAP combination for therapy of gingival carcinoma in vitro. Our treatment modality using cisplatin chemotherapy and CAP treatment may provide the optimal treatment strategy for gingival cancer, i.e., local treatment of CAP against gingival cancer must have a maximized cancer cell-killing capacity and minimized systemic side-effects of cisplatin. The increased expression of tumor-suppressor proteins and apoptosis-related proteins, resulting in apoptotic death of cancer cells, also verified the synergistic anticancer activity of the combination of cisplatin and CAP, as shown in Figure 8.

Even though the maximum anticancer activity was observed for the combination of 5 μM cisplatin plus 3 min of CAP treatment, the viability of HGF-1 cells also decreased, and, thus, the optimal combination of cisplatin/CAP should be determined. From these points of view, appropriate combinations can be selected to maximize anticancer activity with minimized cytotoxicity against normal cells. In our opinion, 1 μM cisplatin plus 3 min of CAP treatment or 3 μM cisplatin plus 1 min of CAP treatment are the preferred combination options rather than other treatment regimens, because the viability of cancer cells was sufficiently inhibited, while the viability of HGF-1 cells remained higher than 60%. Furthermore, local treatment of disease in the periodontal region is required to improve therapeutic efficacy because it is hard to reach the periodontal region with the systemic delivery of drugs [46]. Thus, the local treatment modality should be developed to maximize drug efficacy against the periodontal and gingival region [47,48]. Our finding provides an appropriate treatment strategy of gingival cancer using a local application of CAP with systemic chemotherapy of cisplatin.

## 3. Materials and Methods

### 3.1. Materials

Cisplatin, 3-(4,5-dimethylthiazol-2-yl)-2,5-diphenyltetrazolium bromide (MTT), and 2`,7`-dichlorofluorescein diacetate (DCFH-DA) were purchased from Sigma-Aldrich Chemical Co. (St. Louis, MO, USA). All organic solvents and other chemicals including dimethyl sulfoxide (DMSO) were used as extra-pure grade. The live/dead cell staining kit was purchased from Biovision (Milpitas, CA, USA). Dulbecco’s modified Eagle medium (DMEM) and DMEM nutrient mixture F-12 (DMEM/F12, 1:1) were purchased from Life Tech. Co. (Grand Island, NY, USA). Human gingival fibroblasts (HGF) and SCC-15 human squamous cell carcinoma cell lines were obtained from American Type Culture Collection (ATCC, Manassas, VA 20108, USA).

### 3.2. CAP Apparatus for Treatment of Cells

CAP apparatus P500-SM (Sakikake Co. Ltd., Kyoto, Japan) was used to treat cancer cells, as shown in Figure 1. CAP apparatus is consisted of a gas supply system, a mass flow controller (MFC), a plasma jet and a high-voltage AC power supply, as shown in Figure 1A. This apparatus was employed to produce a stable discharge with a 20 kHz radio frequency at atmospheric conditions, and the discharge power was adjusted at 45 W and at 8.5 kV. From the gas supply system, 5 L/min of Ar gas was supplied for plasma treatments, and the purity of gas was higher than 99.9%. A 12-well plate containing cells was mounted on the XY stage. The distance between the plasma jet nozzle and the media in the 12-well plate was adjusted to 3 cm, as shown in Figure 1B.

### 3.3. OES Measurement in the Plasma Jet Flame

A fiber optic spectral analyzer (Avantes AvaSpec-ULS2048CL-EVO-RS, spectra range from 200 to 1100 nm, Apeldoorn, Netherlands) was used to measure optical emission spectra of the CAP. An optical fiber (Avantes, Fiber-optic cable, FC-UV400-2, Apeldoorn, Netherlands) was located at a distance of 10 mm from the nozzle of the plasma jet.

### 3.4. Cell Culture and CAP Treatment

HGF-1 cells and SCC-15 cells were maintained in DMEM and DMEM/F12 medium, respectively. Each medium was supplemented with 10% heat-inactivated FBS and 1% penicillin/streptomycin at 37 °C in a 5% CO_2_ incubator. To maintain cells, media were exchanged and/or sub-cultured at intervals of every 2–3 days.

For CAP treatment, SCC-15 or HGF-1 cells were seeded in 12 wells (at a density of 1 × 10^5^ cells in each well), and these cells were cultured in the 5% CO_2_ incubator overnight before treatment of CAP. Following this, cells were treated with CAP equipment (Sakigake; Kyoto, Japan, P500-SM, Figure 1A) with 5 kV for various periods of time (10 s–5 min) (flow rate: 5 L/min, Argon gas) at a 3 cm distance from the media (Figure 1B).

### 3.5. Cell Viability Assay

Cell viability was checked with an MTT assay. Cells seeded in 12 wells (1 × 10^5^ cells in each well) were primarily treated with various concentrations of cisplatin. Six hour later, cells were washed with phosphate-buffered saline (PBS, 0.01M, pH 7.4) and replaced with fresh serum-free media. Following this, cells were treated again with CAP. To examine the treatment period, cells were exposed to CAP for various periods of time. Cells were placed in the 5% CO_2_ incubator for 24 h. After that, the MTT solution (500 μg/mL PBS, pH 7.4) was then added to the each well, which was followed by incubation for 2 h in the 5% CO_2_ incubator at 37 °C. The supernatant was discarded and replaced with 100 μL DMSO. The viability of cells was measured absorbance at 560 nm with a microplate reader (EPOCH, BioTek Instruments Inc., Winooski, VT., USA). All the values in the MTT assay are average ± S.D. (standard deviation) from four wells.

### 3.6. ROS Assay

ROS generation by the treatment of cisplatin and CAP was monitored with the DCFH-DA method. Briefly, cells seeded in 12 wells (1 × 10^5^ cells in each well) were treated with cisplatin and/or CAP as described for the MTT assay. Briefly, cells were treated with cisplatin for 6 h, washed with PBS, replaced with fresh serum-free media and then treated with CAP. After that, DCFH-DA (final concentration, 20 μM) in the serum-free media was added and further incubated in the 5% CO_2_ incubator for 1 h. Intracellular ROS was fluorescently measured with an Infinite M200 pro microplate reader (Tecan, Mannedorf, Switzerland) (excitation wavelength: 485 nm, emission wavelength: 520 nm). This procedure was performed under dark conditions. ROS values are average ± S.D. (standard deviation) from four wells.

### 3.7. Live and Dead Cell Staining

To observe the viability behavior of the cells, 1 × 10^5^ cells in 12 wells were treated with cisplatin for 6 h followed by CAP treatment as described above. Then, cells were stained with live/dead cell dye as follows: 1 × 10^5^ cells in 12 wells were treated with cisplatin and/or CAP as described for the MTT assay. Briefly, cells were treated with cisplatin for 6 h followed by CAP treatment. Then, cells were harvested using trypsin–EDTA. The harvested cells were washed with PBS twice and then incubated for 15 min with 1 mL of the live/dead cell staining solution (1 μL live dye (1 mM) and 1 μL of the dead dye solution (1 mg/mL propidium iodide) (PI) in staining buffer, 1 mL) at 37 °C. Following this, cells were observed with a fluorescence microscope under dark conditions (Eclipse Ni-U, Nikon, Tokyo, Japan).

### 3.8. Western Blot Assay

To monitor the expression of apoptotic proteins in cells, SCC-15 cells were treated with cisplatin and CAP as described above. These cells were further incubated for 24 h. After that, cells were harvested from all wells and then washed with PBS twice. These cells were harvested by centrifugation. Cell pellets were lysed with 30 μL RIPA lysis buffer (150 mM NaCl, 1.0% IGEPAL^®^ CA-630, 0.5% sodium deoxycholate, 0.1% SDS, 50 mM Tris, pH 8.0) containing protease and dephosphatase inhibitors (Thermo Fisher Scientific; Waltham, Massachusetts, USA) for 30 min at 4 °C. Following this, the lysed cells were centrifuged at 16,000× *g* for 20 min, and then supernatants were removed. Protein concentration was measured with a BCA protein assay kit at 562 nm (Pierce Biotechnology, IL 61105, USA) using a UV–vis spectrophotometer (Genesis G10S UV-VIS spectrophotometer, Thermo Scientific, USA). Then, 50 μg of protein was loaded to SDS polyacrylamide gel electrophoresis (SDS-PAGE). This was transferred to a polyvinyl difluoride (PVDF) membrane, blocked with 5% skim milk in TBS-T and then probed with a primary antibody such as caspase 9 (1:1000), PTEN (1:1000), P53 (1:1000) or β-actin (1:40,000) (Abcam; Cambridge, UK). Following this, the protein was treated with a secondary antibody (donkey anti rabbit (1:4000), donkey anti mouse (1:10,000) (Thermo Fisher Scientific)) for 1 h. The immunoblots were detected by chemiluminescence and then quantified with digital analyses using the ImageJ software program.

### 3.9. Flow Cytometry for Analysis of Apoptosis/Necrosis of Cancer Cells

The apoptosis and necrosis behavior of cells after cisplatin/CAP treatment was monitored using propidium iodide (PI) for necrosis and FITC–annexin V (sc-4252 FITC, Santa Cruz Biotech., Inc., Dallas, Texas, USA) for apoptosis. SCC-15 cells were treated with cisplatin for 6 h followed by CAP treatment as described above. Cells were further incubated in the 5% CO_2_ incubator at 37 °C for 24 h. Cells were washed with PBS and harvested by trypsinization. Then, collected pellets were resuspended into the binding buffer (10 mM HEPES pH 7.4, 150 mM NaCl, 5 mM KCl, 1 mM MgCl_2_, and 1.8 mM CaCl_2_) containing FITC–annexin V (1 µg/mL) to stain apoptotic cells, which was followed by incubation for 20 min in the 5% CO_2_ incubator at 37 °C. After that, PI (10 µg/mL) was added to this solution to stain the necrotic cells, which were incubated for a further 10 min in the 5% CO_2_ incubator at 37 °C. Following this, the apoptosis/necrosis behavior of cells was analyzed with a FACScan flow cytometer (Becton Dickenson Biosciences, San Jose, CA, USA) equipped with an excitation laser line at 488 nm for FITC–annexin and a 575 ± 15 nm band pass filter for PI. All procedures of apoptosis/necrosis dye staining were performed in dark conditions.

### 3.10. Statistical Analysis

Statistics of the experimental results were evaluated with the Student t-test, and *p* values lower than 0.05 were considered as a value of statistical significance. 

## 4. Conclusions

The synergistic anticancer effect of cisplatin and CAP treatment was examined using SCC-15 and HGF-1 cells in vitro. Viability of SCC-15 cells was inhibited by cisplatin with a dose-dependent manner and/or CAP treatment time. HGF-1 cells also showed decreased viability by treatment with cisplatin and CAP. A combination of 1 μM cisplatin plus 3 min of CAP treatment or 3 μM cisplatin plus 1 min of CAP treatment showed a synergistic anticancer effect with appropriate cytotoxicity against normal cells. ROS generation and dead cell staining were also increased by the increase in CAP treatment time. Furthermore, tumor-suppressor proteins and apoptosis-related enzymes also increased according to the treatment time of CAP. We showed a synergistic effect of cisplatin and CAP treatment against SCC-15 cells with low cytotoxicity against normal cells.

## Figures and Tables

**Figure 1 ijms-21-07646-f001:**
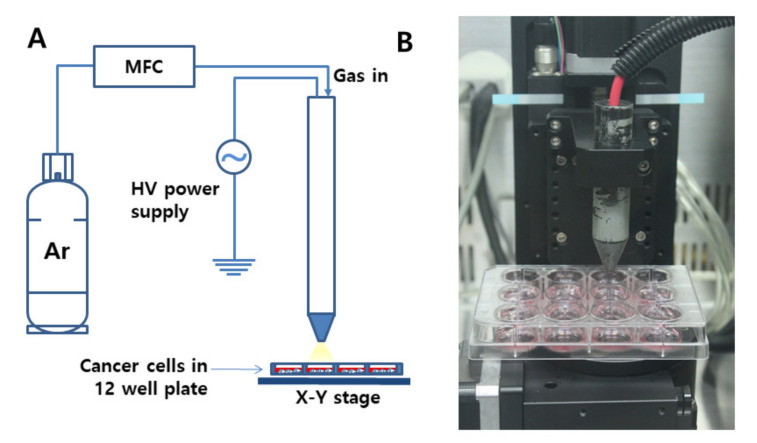
(**A**) Schematic diagram of cold atmospheric plasma (CAP) equipment for the treatment of cells. (**B**) Photograph of the CAP nozzle. Flow rate of Ar gas was 5 L/min. Discharge power was adjusted at 45 W, at 8.5 kV, and 20 kHz radio frequency at atmospheric condition. The distance was 3 cm from the nozzle to the cells in a 12-well plate.

**Figure 2 ijms-21-07646-f002:**
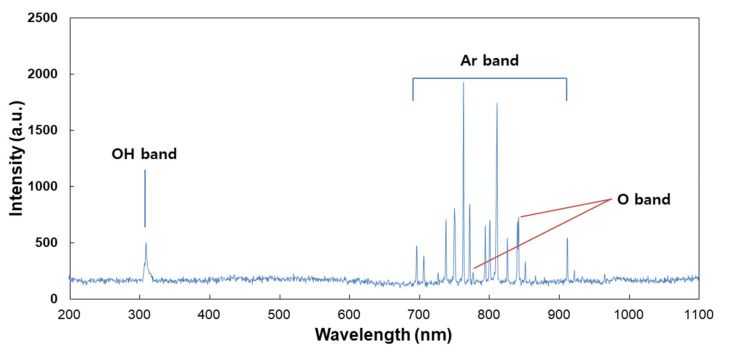
Optical emission spectrum (OES) measurement of the CAP plume. An optical fiber was measured from the position of 3 mm along the axial direction of the CAP plume.

**Figure 3 ijms-21-07646-f003:**
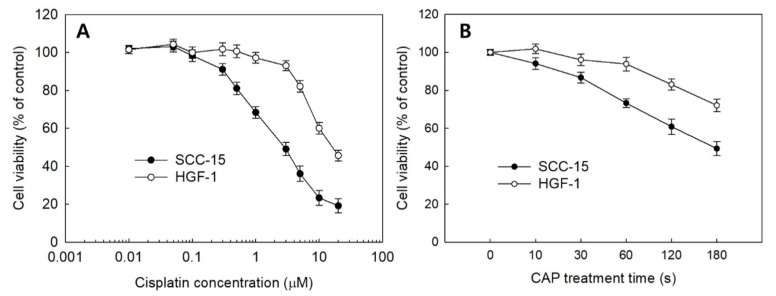
The viability of SCC-15 and HGF-1 cells. (**A**) The effect of cisplatin treatment; (**B**) CAP treatment time.

**Figure 4 ijms-21-07646-f004:**
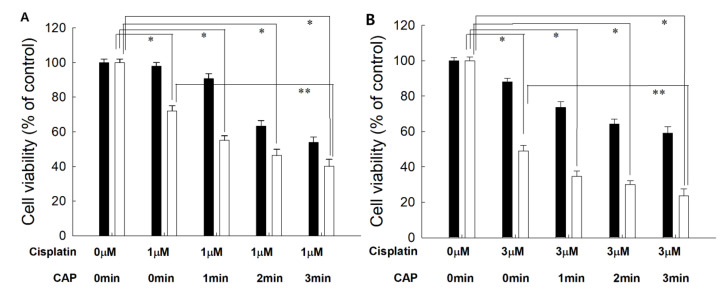
The effect of CAP exposure time on the cell viability of HGF-1 (■) and SCC-15 (□). (**A**) Cisplatin, 1 μM; (**B**) cisplatin, 3 μM. Cisplatin at a dose of 1.0 μM or 3.0 μM was exposed to cells for 6 h, and then media was replaced with fresh media. These were then irradiated with CAP for various periods of time. *, **: *p* < 0.01.

**Figure 5 ijms-21-07646-f005:**
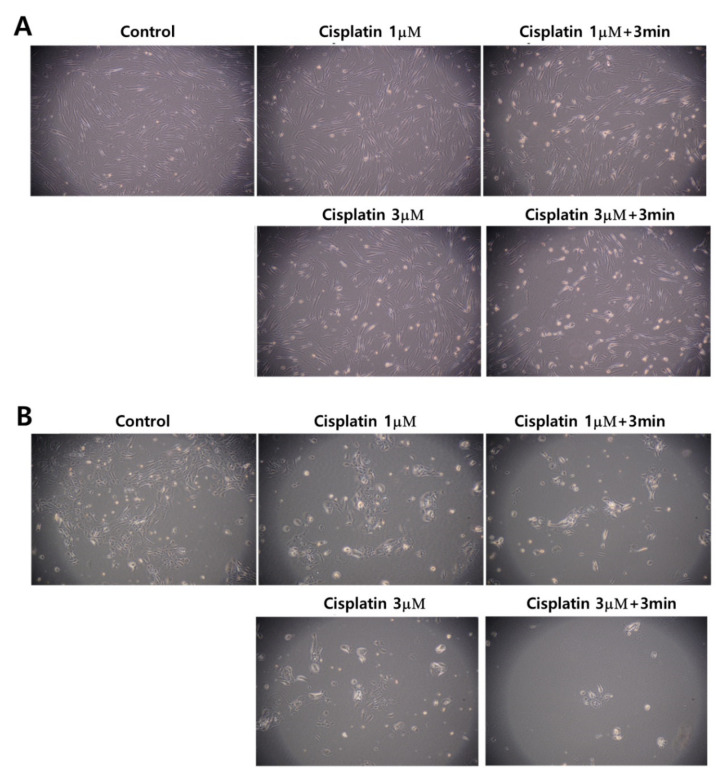
Observation of changes in morphology and density. (**A**) HGF-1 cells; (**B**) SCC-15 cells. The effect of 1 μM and 3 μM cisplatin concentration with 3 min of CAP exposure. (Magnification: 100×)

**Figure 6 ijms-21-07646-f006:**
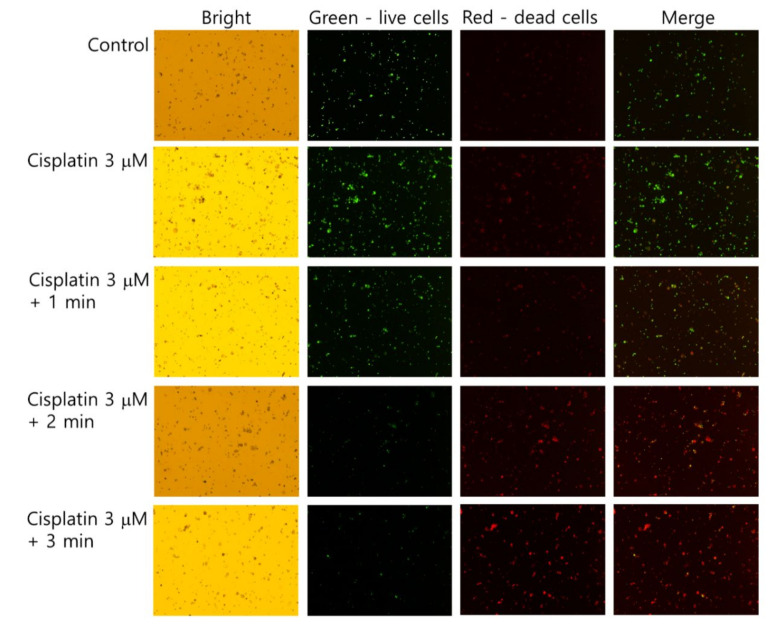
Live and dead cell staining. SCC-15 cells and HGF-1 cells were treated with cisplatin and CAP, as described in Figure 4. Cells treated with cisplatin/CAP were harvested and then stained with a live/dead cell staining solution. Green and red represent live and dead cells, respectively. (Magnification: 40×)

**Figure 7 ijms-21-07646-f007:**
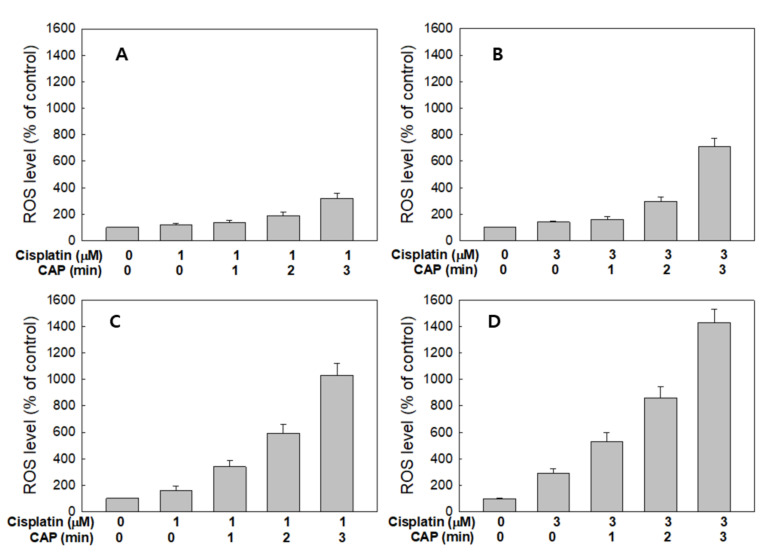
Synergistic combination effect of cisplatin and plasma on reactive oxygen species (ROS) generation. ROS generation of HGF-1 (**A**,**B**) and SCC-15 cells (**C**,**D**) after the CAP exposure (1, 2 and 3 min) with 1 mM (a and c) and 3 μM (b and d) of cisplatin.

**Figure 8 ijms-21-07646-f008:**
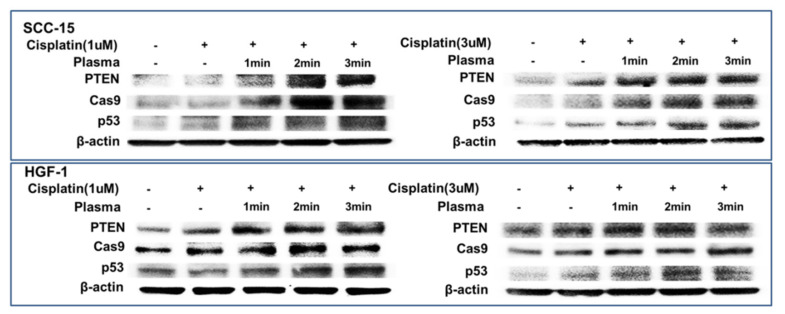
Western blot for the analysis of apoptosis behavior of cells. SCC-15 cells and HGF-1 cells were treated with cisplatin and CAP as described in Figure 4. Cells treated with cisplatin/CAP were then used to analyze tumor suppressor proteins and apoptosis proteins using Western blotting.

**Figure 9 ijms-21-07646-f009:**
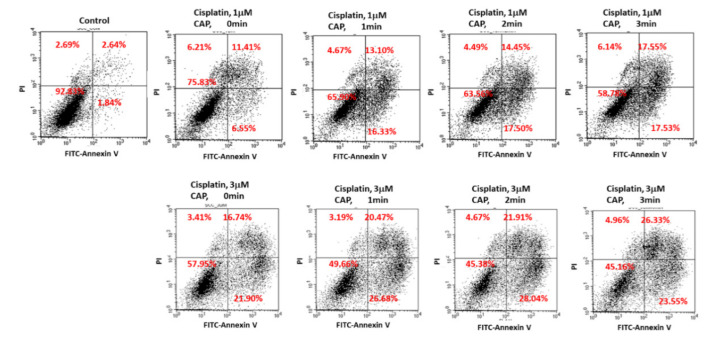
Flow cytometry of apoptosis/necrosis analysis of SCC-15 cells. SCC-15 cells treated with cisplatin/CAP as described in Figure 4 were then stained with fluorescein isothiocyanate (FITC)–annexin V for apoptosis and propidium iodide (PI) for necrosis. Apoptosis/necrosis analysis of SCC-15 cells using flow cytometry under CAP exposure and cisplatin.

**Table 1 ijms-21-07646-t001:** IC_50_ of cisplatin and CAP treatment against SCC-15 cells.

	IC_50_	
	HGF Cells	SCC-15 Cells
Cisplatin	16.0 μM	2.8 μM
CAP treatment	> 300 s	173 s

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
