# Peer review of "Combinatorial Effect of Cold Atmosphere Plasma (CAP) and the Anticancer Drug Cisplatin on Oral Squamous Cell Cancer Therapy"

_ijms, 2020, doi:10.3390/ijms21207646_

Round 1

Reviewer 1 Report

The authors investigated the effect of combination of Cold atmospheric plasma (CAP) and cisplatin on cell growth and cell death of the oral squamous cell carcinoma (OSCC) cell line SCC-15 in vitro. The results demonstrated the viability of SCC-15 cells was inhibited by cisplatin with dose-dependent manner and CAP treatment time. Combination of CAP and cisplatin significantly induced ROS production and cell death. Overall, the aim of this study is clear and the data were well-organized and presented. Regarding this manuscript, I have a few concerns: 

1) How did you discriminate "synergistic effect" not "additive effect" of the combination of CAP and cisplatin?

2) For Figure 5, can you make a statistic on live and dead cells?

3) For Figure 7, the authors are trying to detect the markers of apoptosis by using Western blot. One of the key markers is caspase 3, which is cleaved during cell apoptosis. Can you add the Western blot results to check caspase 3? which is better than PTEN and p53. And them make a statistics for the western blot results.

4) Change "IC50" to "IC50" in the text.

Author Response

Response to reviewer 1’s comment

The authors investigated the effect of combination of Cold atmospheric plasma (CAP) and cisplatin on cell growth and cell death of the oral squamous cell carcinoma (OSCC) cell line SCC-15 in vitro. The results demonstrated the viability of SCC-15 cells was inhibited by cisplatin with dose-dependent manner and CAP treatment time. Combination of CAP and cisplatin significantly induced ROS production and cell death. Overall, the aim of this study is clear and the data were well-organized and presented. Regarding this manuscript, I have a few concerns: 

1) How did you discriminate "synergistic effect" not "additive effect" of the combination of CAP and cisplatin?

Answer) Thanks for your comment. At this moment, we discriminate that “synergistic effect” shows higher anticancer effect by combination of two kinds of method than simply sum of cytotoxicity of them. Furthermore, they should be showed lower cytotoxicity against normal cells with suitable anticancer activity. Practically, we showed that combination of cisplatin and CAP treatment has synergistic anticancer activity with low cytotoxicity against normal cells. Anyway, the term of “synergistic” for this case is controversial. Therefore, we changed the title from “Synergistic effect of cold atmosphere plasma (CAP) and anticancer drug on oral squamous cell cancer therapy” to “Combinatorial effect of cold atmosphere plasma (CAP) and anticancer drug on oral squamous cell cancer therapy” to avoid controversial issues.

2) For Figure 5, can you make a statistic on live and dead cells?

Answer) Thanks for your comment. At this moment, we did not calculate and specify the statistic of live/dead cells from this Figure. In this figure, we just showed the images of live/dead cell differences from Figure 3(b). Anyway, we revised the Figure 3 and added statistical value. Furthermore, we discussed more for Figure 5 because this experiment derived from Figure 3(b).

Figure 3. The effect of CAP exposure on the cell viability of HGF-1 (â– ) and SCC-15 (â–¡). (a) cisplatin, 1 mM; (b) cisplatin, 3 mM. 1.0 mM or 3.0 mM  cisplatin was exposed to cells for 6h and then media was replaced with fresh media. These were then irradiated with CAP for various period of time. *,** : p < 0.01.

Live/dead cell staining of SCC-15 cells supported the results of Figure 3 and 4. As shown in Figure 5, dead cells having red color was gradually increased at combination of cisplatin and CAP treatment compared to control. The higher the treatment time of CAP significantly decreased live cells and then dead cells simultaneously increased compared to control or cisplatin only. Otherwise, live cell staining in HGF-1 cells having green color was relatively more than dead cell staining (Figure s2), indicating that HGF-1 cells were relatively influenced less than SCC-15 cells. Figure 3, 4 and 5 showed that combination of cisplatin and CAP treatment induces synergistic death of cancer cells with alleviated cytotoxicity against normal cells.

3.10. Statistical analysis

Statistics of the experimental results were evaluated with the Student t-test and then considered p value lower than 0.05 as a significant value statistically.

3) For Figure 7, the authors are trying to detect the markers of apoptosis by using Western blot. One of the key markers is caspase 3, which is cleaved during cell apoptosis. Can you add the Western blot results to check caspase 3? which is better than PTEN and p53. And them make a statistics for the western blot results.

Answer) Thanks for your comment. Western blot results in Figure 7 just supported the relative ratio of live/dead cells from the results of Figure 3~6. According to your comment, caspase 3 is also good candidate to observe apoptosis of cells. However, Caspase-9 is also strong evidence of apoptosis of cancer cells. Cleaved Caspase-9 (Cas-9) was increased according to the increase of CAP treatment time and this result indicated that apoptosis of cells has been occurred with proportion to CAP treatment time. Anyway, we discussed more about this result in the Results and discussion section.

4) Change "IC50" to "IC50" in the text.

Answer) Thanks to your comment. According to your comment, this word corrected.

Table 1. IC50 of cisplatin and CAP treatment against SCC-15 cells.

IC50

HGF cells

SCC-15 cells

Cisplatin

CAP treatment

16.0 mM

> 300 s

2.8 mM

173 s

Reviewer 2 Report

My opinion is that the work is not quite well organised, and some statements of the authors need further proofs.

First, I would like to recommend the authors to completely edit the text as there are typos and punctuation errors: for example, Line 132 “were efficiently responded”…; Line 124 “Figure 3~8”; Line 126: “SCC-15 cells were efficiently 126 responded to cisplatin concentration until 10 µM” there is no meaning in the sentences…please revise; Line 147-148:” HGF-1 cells are still well-tolerated to the exposure of cisplatin and 147 CAP while the viability of SCC-15 cells was significantly decreased, i.e. viability of SCC-15 cells were less 148 than 30% at 3 microM cisplatin plus 3 min CAP while the viability of HGF-1 cells was higher than 50%..” etc.

The single treatment results are not described well: the authors show cytotoxicity of cisplatin only on SCC-15 cells and results are shown in Suppl. These are important results and they need to be performed on the other cell line too. Moreover, the comparison should be given as a regular figure, not a supplementary material. The authors compare the estimated IC50 of cisplatin and CAP for the two cell lines, stating that the difference was slight, which is not true as for HGF cells the concentration is almost 5 times difference.

The dual treatment results lack adequate explanation. There are many mistakes in the sense of grammar and spelling which prevent the reader from clarity and understanding. Comparative statements for some results are underestimated or overrated. The results for cell morphology after treatment are faintly elucidated. The micrographs of Fig. 4 lack quality and clarity. Fig. 5 contains micrographs taken at different levels of light intensity. Moreover, it contains treatment protocols for which no cytotoxicity results are shown (consider 3 µM cisplatin and 2 min CAP. What is the difference in Figure 3 and Figure 4 a and c? Specify it!

Results with the expression of caspase 9 and p53 need further proof from RT-qPCR. The authors need to show what is going on the level of mRNA too. The Western blot results are shown in Fig. 7 are taken under different levels of light enhancement. Moreover, Western blotting is a semi-quantitative method which under evaluates the results for any changes in gene expression under certain treatment conditions. The bands are cut in a way that allows the reader to doubt the results.

FACS analysis for apoptosis/necrosis evaluation on Figure 8 has drawbacks too. First, results are only for SCC-15 cells. No results for the other cell culture are presented. The dot-plot diagrams complement the quantitated data in Fig. 8B. No need for the two presentations. The authors could give the dot-plots with the % of cells in each of the quadrants, representing different stages and types of cell death. No STDVs are visible. These are data from one experiment only? The authors need a positive control for apoptosis and necrosis for data evaluation and quantitation.

Results and Discussion is a good option for paper. Here, the authors need to further expand the Discussion part as until the last results with FACS and cell death they do not discuss the obtained results enough and this diminishes the quality of the work.

Author Response

Response to reviewer 2’s comment

My opinion is that the work is not quite well organised, and some statements of the authors need further proofs.

First, I would like to recommend the authors to completely edit the text as there are typos and punctuation errors: for example, Line 132 “were efficiently responded”…; Line 124 “Figure 3~8”; Line 126: “SCC-15 cells were efficiently 126 responded to cisplatin concentration until 10 µM” there is no meaning in the sentences…please revise; Line 147-148:” HGF-1 cells are still well-tolerated to the exposure of cisplatin and 147 CAP while the viability of SCC-15 cells was significantly decreased, i.e. viability of SCC-15 cells were less 148 than 30% at 3 microM cisplatin plus 3 min CAP while the viability of HGF-1 cells was higher than 50%..” etc.

Answer) Thanks to your comment. According to your comment, we corrected some of typical errors and fully revised as follows:

From the results of viability of cells, optimal combination of cisplatin concentration and CAP treatment time was studied. As shown in Figure 3 and 4, 1 mM or 3 mM of cisplatin was combined with various treatment time of CAP to find maximal anticancer activity against cancer cells and minimal cytotoxicity against normal cells. Practically, higher than 80% of HGF-1 cells were viable until 5.0 mM. As shown in Figure 3, cells responded to CAP treatment time and viability was gradually decrease according to the increase of treatment time when various treatment time of CAP were combined with 1 mM or 3 mM cisplatin. Even though viability of HGF-1 cells was also decreased according to the treatment time of CAP, their viability was higher than SCC-15 cells at all treatment regimens. When 1 min treatment of CAP was combined with 1 mM cisplatin, less than 60% in SCC15 cells was viable while higher than 80% of HGF-1 cells were viable. 1 min treatment of CAP combined with 3 mM cisplatin induced decrease of SCC-15 cell viability less than 40% while higher than 70% of HGF-1 cells were viable at same treatment modality. Furthermore, viability of HGF-1 cells at 3 mM cisplatin was not significantly changed compared to the results of 1 mM cisplatin while viability of SCC-15 cells were significantly decreased at 3 mM cisplatin compared to 1 mM cisplatin. These results indicated that SCC-15 cells are more sensitive to cisplatin treatment compared to HGF-1 cells. Even though higher dose of cisplatin in combination with CAP resulted in stronger antitumor activity, lower dose of cisplatin combined with longer treatment time of CAP is preferred for cancer treatment because CAP brings only about local irritations in the normal tissues while cisplatin induces systemic side-effects against human body.

The single treatment results are not described well: the authors show cytotoxicity of cisplatin only on SCC-15 cells and results are shown in Suppl. These are important results and they need to be performed on the other cell line too. Moreover, the comparison should be given as a regular figure, not a supplementary material. The authors compare the estimated IC50 of cisplatin and CAP for the two cell lines, stating that the difference was slight, which is not true as for HGF cells the concentration is almost 5 times difference.

Answer) Thanks to your comment. According to your comment, we moved the supplementary figure to main manuscript as Figure 3. Furthermore, we removed “slightly” because IC50 of cisplatin and CAP were significantly different between SCC-15 cells and HGF-1 cells.

  In this study, we just showed typical study for the effect of CAP treatment effect against cancer cells when combined with cisplatin. And we focused what the optimal condition is for the treatment modality of CAP/cisplatin combination against malignant SCC-15 cells and non-malignant HGF-1 cells. We will study more with various cancer cells and report in other report.

The effect of sole treatment of cisplatin or CAP against SCC-15 cells and HGF cells was examined in vitro as shown in Figure 3. Viability of SCC-15 cells and HGF cells was gradually decreased according to the increase of cisplatin concentration and CAP treatment time. Interestingly, SCC-15 cells showed lower viability against cisplatin treatment compared to HGF-1 cells as shown in Figure 3(a). That is, viability of SCC-15 cells were lower than 50 % 3 mM cisplatin while HGF-1 cells showed higher than 90 % in cell viability. Also, SCC-15 cells resulted in lower viability by CAP treatment compared to HGF-1 cells (Figure 3(b)). As shown in Figure 3(b), viability of SCC-15 cells was less than 50 % at 3 min (180 s) treatment while higher than 70 % HGF-1 cells were viable. These results indicated that HGF-1 cells were properly tolerated against treatment of cisplatin or CAP rather than SCC-15 cells. As shown in Table 1, the IC50 value of HGF-1 and SCC-15 cells by treatment of cisplatin was 16.0 mM and 2.8 mM. Furthermore, viability of SCC-15 cells was also gradually decreased according to the treatment time against CAP. IC50 value of HGF-1 and SCC-15 cells by treatment of CAP was approximately higher than 300 s and 173 s, respectively. Interestingly, IC50 value of HGF-1 cells was higher than SCC-15 cells.

Figure 3. The effect of cisplatin treatment (a) and CAP treatment time (b) on the viability of SCC-15 and HGF-1 cells.

The dual treatment results lack adequate explanation. There are many mistakes in the sense of grammar and spelling which prevent the reader from clarity and understanding. Comparative statements for some results are underestimated or overrated. The results for cell morphology after treatment are faintly elucidated. The micrographs of Fig. 4 lack quality and clarity. Fig. 5 contains micrographs taken at different levels of light intensity. Moreover, it contains treatment protocols for which no cytotoxicity results are shown (consider 3 µM cisplatin and 2 min CAP. What is the difference in Figure 3 and Figure 4 a and c? Specify it!

Answer) Thanks for your comment. According to your comment, we fully revised the manuscript, corrected grammatical error and rewritten some of the results.

 Since we indicated that longer treatment time of CAP is preferred for cancer therapy, we wanted to test the effect of various concentration of cisplatin at 3min CAP treatment (in Figure 4 (Figure 5 in revised manuscript)). However, at this moment, the results in Figure 4(a) and (c) (Figure 5(a) and (c) in revised manuscript) should be removed to avoid complexity with Figure 3 (Figure 4 in revised manuscript).

Micrographs in Figure 4 (Figure 5 in revised manuscript) just show the changes of density of cell number to support Figure 3 (Figure 4 in revised manuscript) according to the changes of CAP treatment time and cisplatin concentration. Anyway, we revised it to support results in Figure 3 (Figure 4 in revised manuscript).

Figure 5 also supported the results of Figure 3 and 4. Even though morphology and density of HGF-1 cells were changed at 3mM cisplatin plus 3 min CAP treatment time, cell density and morphology of HGF-1 cells was relatively maintained at 1mM cisplatin, 1mM cisplatin plus 3 min CAP treatment time and 3mM cisplatin as shown in Figure 5(a). Otherwise, SCC-15 cells showed greater changes in morphology and cell density by treatment of cisplatin and/or CAP. At 3mM cisplatin plus 3 min CAP treatment time, cell density and morphology of SCC-15 cells were significantly decreased and changed as shown in Figure 5(b). These results supported the results of Figure 4.

Figure 5. Observation of changes of morphology and density of HGF-1 cells (a) and SCC-15 (b) cells. The effect of 1mM and 3 mM cisplatin concentration with 3 min CAP exposure time.

Results with the expression of caspase 9 and p53 need further proof from RT-qPCR. The authors need to show what is going on the level of mRNA too. The Western blot results are shown in Fig. 7 are taken under different levels of light enhancement. Moreover, Western blotting is a semi-quantitative method which under evaluates the results for any changes in gene expression under certain treatment conditions. The bands are cut in a way that allows the reader to doubt the results.

Answer) Thanks for your comment. At this moment, we showed wester blotting results to support Figure 4 and 5. As shown in Figure 7 (Figure 8 in revised manuscript), western blot clearly showed that apoptosis-protein expression of SCC-15 cells was significantly increased compared to HGF-1 cells. Anyway, we discussed more for this result in the results and discussion section.

Figure 8 shows the expression of apoptosis-related signals such as PTEN, cleaved caspase-9 (Cas-9) and p53 proteins of SCC-15 cells and HGF-1 cells. As shown in Figure 8, tumor-suppressor protein, PTEN, was gradually increased both SCC-15 cells and HGF-1 cells, indicating that CAP treatment increases anti-tumor activity with expression of tumor-suppressor genes. Furthermore, apoptosis-related enzyme, Cas-9, in SCC-15 cells was remarkably changed according to the CAP treatment both of 1 mM and 3 mM cisplatin, i.e. the higher the treatment time of CAP gradually increases expression intensity of Cas-9. Otherwise, the changes of Cas-9 expression intensity in HGF-1 cells were relatively not significant compared to SCC-15 cells. The expression intensity of p53 protein was also gradually increased by treatment with CAP both SCC-15 cells and HGF-1 cells. These results indicated that CAP treatment in combination with cisplatin induces apoptosis of tumor cells and increases tumor-suppressor protein, resulting in increase of antitumor activity. Figure 9 shows the flow cytometric analysis of apoptosis/necrosis of SCC-15 cells. As expected, apoptosis/necrosis of SCC-15 cells was gradually increased according to the increase of treatment time of CAP. Furthermore, early/late apoptosis of cancer cells at 3 mM of cisplatin was higher than 1 mM of cisplatin. Saadati et al., also reported that apoptosis-related protein expression such as BAX/BCL-2 was significantly changed in B16F10 melanoma cells by indirect/direct treatment with CAP [43]. They were reported that CAP treatment against B16F10 melanoma cells greatly inhibited viability of malignant cells in vitro and in vivo through apoptosis of cells.

  1. Saadati, F.; Mahdikia, H.; Abbaszadeh, H.A.; Abdollahifar, M.A.; Khoramgah, M.S.;Shokri, B. Comparison of direct and indirect cold atmospheric-pressure plasma methods in the B16F10 melanoma cancer cells treatment. Rep. 2018, 8, 7689.

FACS analysis for apoptosis/necrosis evaluation on Figure 8 has drawbacks too. First, results are only for SCC-15 cells. No results for the other cell culture are presented. The dot-plot diagrams complement the quantitated data in Fig. 8B. No need for the two presentations. The authors could give the dot-plots with the % of cells in each of the quadrants, representing different stages and types of cell death. No STDVs are visible. These are data from one experiment only? The authors need a positive control for apoptosis and necrosis for data evaluation and quantitation.

Answer) Thanks for your comment. Practically, the FACS results were from one experiments from cisplatin/CAP treatment. According to your comments, we revised the Figure 8 (Figure 9 in revised manuscript) and give the dot-plots with the % of cells in each quadrants. Figure 9 is to support and to approve synergistic cell death by CAP/cisplatin treatment in Figure 4.

Figure 9. Flow cytometry of apoptosis/necrosis analysis of SCC-15 cells. SCC-15 cells treated with cisplatin/CAP as described in Figure 4 were then stained with FITC-Annexin V for apoptosis and PI for necrosis. Apoptosis/necrosis analysis of SCC-15 cells using flow cytometry under the CAP exposure and cisplatin.

Results and Discussion is a good option for paper. Here, the authors need to further expand the Discussion part as until the last results with FACS and cell death they do not discuss the obtained results enough and this diminishes the quality of the work.

Answer) Thanks for your comment. According to your comment, we discussed more for the results of FACS and cell death and fully revised the manuscript. Practically, Figure 7 and 8 (Figure 8 and 9 in the revised manuscript) is to support and approve the synergistic anticancer results in Figure 4. Anyway, we discussed more about the apoptotic cell death with apoptosis-related proteins in Results and discussion section as follows.

Furthermore, apoptosis-related enzyme, Cas-9, in SCC-15 cells was remarkably changed according to the CAP treatment both of 1 mM and 3 mM cisplatin, i.e. the higher the treatment time of CAP gradually increases expression intensity of Cas-9. Otherwise, the changes of Cas-9 expression intensity in HGF-1 cells were relatively not significant compared to SCC-15 cells. The expression intensity of p53 protein was also gradually increased by treatment with CAP both SCC-15 cells and HGF-1 cells. These results indicated that CAP treatment in combination with cisplatin induces apoptosis of tumor cells and increases tumor-suppressor protein, resulting in increase of antitumor activity. Figure 9 shows the flow cytometric analysis of apoptosis/necrosis of SCC-15 cells. Apoptosis/necrosis of SCC-15 cells was gradually increased according to the increase of treatment time of CAP. Furthermore, early/late apoptosis of cancer cells at 3 mM of cisplatin combined with CAP treatment was higher than 1 mM of cisplatin. Especially, late apoptosis was relatively higher at 3 min CAP treatment while early apoptosis was evidently higher than late apoptosis at less than 2 min CAP treatment. At sole treatment of cisplatin, early apoptosis was dominant at 3 mM of cisplatin rather than late apoptosis while its value at 1 mM of cisplatin was lower than late apoptosis. These results might be due to that SCC-15 cells were responded to cisplatin treatment during initial 6 h and then apoptosis/necrosis was accelerated following cell death.Xu et al., also reported that CAP induces apoptosis/necrosis of multiple myeloma and increased caspase-9 activity [43]. Saadati et al., also reported that apoptosis-related protein expression such as BAX/BCL-2 was significantly changed in B16F10 melanoma cells by indirect/direct treatment with CAP [44]. They were reported that CAP treatment against B16F10 melanoma cells greatly inhibited viability of malignant cells in vitro and in vivo through apoptosis of cells.

  • Xu, X.; Xu, Y.; Cui, Q.; Liu, D.; Liu, Z.; Wang, X.; Yang, Y.; Feng, M.; Liang, R.; Chen, H.; Ye, K.; Kong, M.G. Cold atmospheric plasma as a potential tool for multiple myeloma treatment. Oncotarget. 2018, 9, 18002.
  • Saadati, F.; Mahdikia, H.; Abbaszadeh, H.A.; Abdollahifar, M.A.; Khoramgah, M.S.;Shokri, B. Comparison of direct and indirect cold atmospheric-pressure plasma methods in the B16F10 melanoma cancer cells treatment. Rep. 2018, 8, 7689.

Round 2

Reviewer 2 Report

The revised version of the manuscript by Lee et al. shows improvement following all comments from the reviewers. Though yet needs further revisions.

Here, I shall address some further edits that are necessary for successful completion and publishing in IJMS.

First, the title needs an edit. “Combinatorial effect of cold atmosphere plasma (CAP) and anticancer drug on oral squamous cell cancer therapy”, please specify here the drug –cisplatin. Otherwise, the title sounds too general and may lead to the conclusion that many anticancer drugs will be tested.

The title I suggest is: “Combinatorial effect of cold atmosphere plasma (CAP) and the anticancer drug cisplatin on oral squamous cell cancer therapy”

Line 12: change add “s” to the “biological systems”.

Moreover, many typos yet exist. For example line 12: “In this study, we examined the synergistic effect of”, change with “In this study, we examined the synergistic effect of..”

What I suggest are the authors to use grammar and spelling software programmes to edit and polish the overall expression in English of their work.

Line 119: Write Figure 3B instead of (Figure 3(b)). Everywhere change Figures a or b with capital letters A and B, etc.

I agree with the added text in point 2.2. and all other parts.

Arrange in Figure 6 the levels of image improvement: set brightness and clarity at the same levels, otherwise control and Cisplatin 3 M+3 min are seen blurred and one can think it as image manipulation.

My overall evaluation of this revised version is very good and I suggest the editor accept it for publication after edit of the whole text and after precise edit of all micrographs.

Author Response

The revised version of the manuscript by Lee et al. shows improvement following all comments from the reviewers. Though yet needs further revisions.

Here, I shall address some further edits that are necessary for successful completion and publishing in IJMS.

First, the title needs an edit. “Combinatorial effect of cold atmosphere plasma (CAP) and anticancer drug on oral squamous cell cancer therapy”, please specify here the drug –cisplatin. Otherwise, the title sounds too general and may lead to the conclusion that many anticancer drugs will be tested.

The title I suggest is: “Combinatorial effect of cold atmosphere plasma (CAP) and the anticancer drug cisplatin on oral squamous cell cancer therapy”

Answer) Thanks for your kind comment. According to your comment, we changed the title to “Combinatorial effect of cold atmosphere plasma (CAP) and the anticancer drug cisplatin on oral squamous cell cancer therapy”.

Line 12: change add “s” to the “biological systems”.

Answer) Thanks for your kind comment. According to your comment, we changed the word to biological systems”.

Moreover, many typos yet exist. For example line 12: “In this study, we examined the synergistic effect of”, change with “In this study, we examined the synergistic effect of..”

What I suggest are the authors to use grammar and spelling software programmes to edit and polish the overall expression in English of their work.

Answer) Thanks for your kind comment. According to your comment, we revised the manuscript and corrected the typos. We expressed them as red colors.

Line 119: Write Figure 3B instead of (Figure 3(b)). Everywhere change Figures a or b with capital letters A and B, etc.

Answer) Thanks for your kind comment. According to your comment, we changed all a,b to capital letters A,B. Thanks.

I agree with the added text in point 2.2. and all other parts.

Answer) Thanks for your comments. Your comments helped to improve our manuscript. Thanks.

Arrange in Figure 6 the levels of image improvement: set brightness and clarity at the same levels, otherwise control and Cisplatin 3 M+3 min are seen blurred and one can think it as image manipulation.

Answer) Thanks for your comments. According to your comments, we manipulated image contrast and revised the Figure 6.

My overall evaluation of this revised version is very good and I suggest the editor accept it for publication after edit of the whole text and after precise edit of all micrographs.

Answer) Thanks for your comments. Your comments helped to improve our manuscript. Thanks.
